# STP: Smart Token Pruning for Vision-Language Models: Balancing Importance and Diversity

## Abstract

Large Vision-Language Models (LVLMs) have achieved remarkable success in multimodal reasoning by jointly processing visual and textual information. However, efficient inference in practical applications remains challenging due to the substantial computational and memory overhead of LVLMs. Existing token pruning strategies often face a trade-off: they either prioritize token importance while neglecting semantic diversity, or enforce diversity at the expense of critical tokens. To overcome this limitation, we propose **STP (Smart Token Pruning)**, a novel framework that balances both objectives. We formulate token pruning as a bi-criteria optimization problem that jointly maximizes *semantic diversity*, to preserve broad coverage of visual concepts, and *token importance*, quantified via a new gradient-based saliency score that integrates feature sensitivity and activation strength. STP introduces a unified token selection strategy that adaptively prunes tokens based on their joint diversity-importance score, ensuring both efficient computation and reliable visual-textual reasoning. Extensive experiments across 11 diverse benchmarks show that STP achieves significant reductions in computation and memory usage while maintaining competitive accuracy. This enables scalable and resource-efficient deployment of LVLMs.

## 1 Introduction

In recent years, Large Vision-Language Models (LVLMs) Liu et al. (2023a); Zhu et al. (2023) have achieved remarkable success in multimodal understanding and reasoning by jointly processing textual and visual inputs. Typically, these models tokenise an input image into a large set of visual tokens that are concatenated with textual tokens and fed into a Large Language Model (LLM). This approach enables powerful cross-modal interactions, allowing the model to generate rich and contextually grounded outputs. However, the number of visual tokens, often much larger than textual tokens, leads to significant computational overhead and memory consumption Choromanski et al. (2020); Katharopoulos et al. (2020). Consequently, the increased inference cost hinders the deployment of LVLMs in resource-constrained environments and limits their scalability to high-resolution or video inputs. To this end, token pruning strategies that can reduce the number of unnecessary visual tokens during inference time without degrading the performance of the LVLM are essential to enable practical and scalable LVLM applications.

Existing token pruning methods generally focus on selecting tokens solely on their importance Chen et al. (2024); Shang et al. (2024) or maximizing diversity Alvar et al. (2025) among selected tokens. Methods emphasizing importance tend to overlook semantic redundancy, leading to the retention of visually similar tokens that provide little additional information. Conversely, approaches prioritizing diversity may discard tokens that are crucial for accurate model predictions or fail to capture salient features necessary for downstream tasks. Moreover, many existing strategies Cai et al. (2024); Lin et al. (2025) rely on fixed or heuristic selection criteria that do not dynamically adapt to the internal state of the underlying model or token-level sensitivities. Such rigidity often results in suboptimal pruning performance, where redundant tokens persist or important visual cues are lost, causing degraded accuracy and inconsistent multimodal reasoning. This limitation becomes increasingly pronounced as LVLMs scale to higher resolutions and more complex prompts, where the interaction between visual and textual modalities amplifies the cost of processing redundant tokens. Addition-

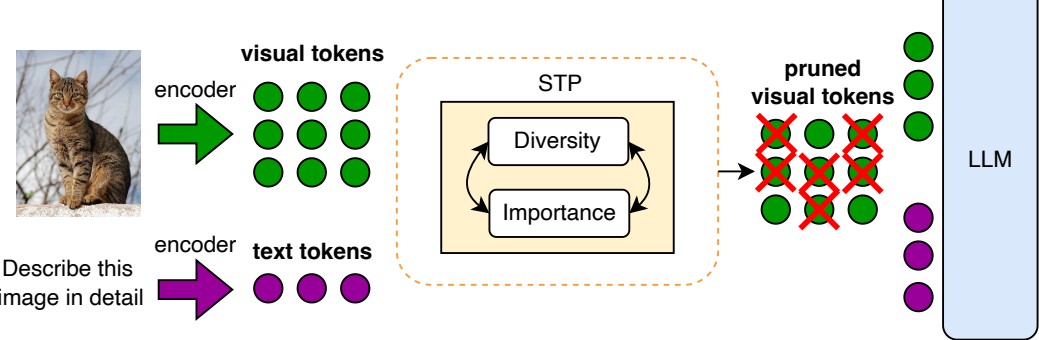

Figure 1: Overview of the STP framework for Large Vision-Language Models (LVLMs). Our method jointly optimizes diversity and importance of visual tokens through a novel bi-criteria optimization strategy, producing a compact yet semantically rich subset of visual tokens. The pruned tokens are then fed into the LLM alongside textual tokens, leading to faster inference and lower memory usage while preserving or improving multimodal reasoning performance.

ally, most prior methods operate under static assumptions about token relevance, failing to leverage model feedback signals (e.g., gradients or attention dynamics) that could guide more informed pruning decisions. To this end, effectively balancing token importance and diversity, while adapting to the evolving computational pathways of LVLMs, remains a fundamental yet unresolved challenge in the design of robust and efficient token pruning.

To address these challenges, We propose a novel token-pruning method that jointly optimizes diversity and importance to select a compact yet semantically representative subset of visual tokens. Framing pruning as a bi-criteria optimization, our approach balances selecting diverse tokens with those most influential to the model's internal representations. As shown in Figure 1, we introduce an importance score based on feature sensitivity and activation strength, combined with a Max–Min diversity strategy to ensure broad semantic coverage. A flexible weighting scheme adapts to different application needs and inputs, while an efficient greedy algorithm scales to large token sets without sacrificing selection quality. Our model-agnostic framework integrates seamlessly with LVLMs, supports adaptive pruning ratios based on image complexity and task requirements, and preserves critical information in challenging scenarios. We provide a lightweight implementation with negligible overhead, achieving faster inference and lower memory usage while maintaining or improving multimodal task accuracy across diverse benchmarks.

Our contributions are summarised as follows:

- We introduce **STP**, a unified and model-agnostic token pruning framework for LVLMs that jointly optimizes semantic diversity and token importance.

- We propose a novel *gradient-based importance scoring* that integrates feature sensitivity and activation strength with a Max-Min coverage strategy, enabling adaptive and context-aware token selection without the need for retraining or architectural modifications.

- Through comprehensive evaluations on 11 multimodal benchmarks, we show that STP achieves significant reductions in computational cost and memory usage while preserving or even improving task performance.

## 2 RELATED WORK

**Large Vision-Language Models:** Large Vision-Language Models (LVLMs) such as BLIP-2 Li et al. (2023b), InstructBLIP Dai et al. (2023), MiniGPT-4 Zhu et al. (2023), LLaVA Liu et al. (2023b), mPLUG-Owl2 Ye et al. (2024), and Qwen-VL Bai et al. (2023) have demonstrated strong multimodal reasoning capabilities. BLIP-2 uses a Query Transformer to extract informative visual tokens from a frozen encoder, enabling efficient alignment with a frozen LLM. InstructBLIP extends this with instruction tuning, while MiniGPT-4 and LLaVA rely on linear projections and weak supervision, risking semantic misalignment. More recent models like mPLUG-Owl2 and Qwen-VL intro-

duce adaptive fusion and broader instruction tuning to improve grounding. Despite these advances, hallucination remains a challenge due to limited grounding, frozen backbones, and next-token training objectives Li et al. (2023c); Rohrbach et al. (2018). Benchmarks such as CHAIR Rohrbach et al. (2018), POPE Li et al. (2023c), and MMHalBench Sun et al. (2023) highlight persistent grounding failures. Recent efforts like LLaVA-RLHF Sun et al. (2023) demonstrate that reinforcement learning can improve factual alignment.

**Token Pruning:** Advanced computer vision models increasingly rely on computationally intensive transformer architectures Vaswani et al. (2017). To mitigate this, token pruning has emerged as a key optimization strategy, enhancing inference efficiency by dynamically selecting a subset of informative tokens. Initial work on Vision Transformers (ViTs) Dosovitskiy et al. (2021), such as DynamicViT Rao et al. (2021) and SPViT Heo et al. (2022), focused on adaptive selection and sparsification to discard redundant tokens. This concept was further refined by methods like Token-Learner Ryoo et al. (2021) and ToMe Pilanci et al. (2022), which learn to merge or drop tokens based on their importance or similarity. More recent approaches minimize the performance degradation associated with pruning. For instance, some techniques fuse information from pruned tokens back into the retained ones to preserve valuable context Wei et al. (2023). Another state-of-the-art method uses a fast post-training framework with dynamic programming to reduce FLOPs without accuracy loss across diverse architectures, including CNNs and ViTs Xu et al. (2025). The principle extends beyond static images, with methods like ADAPTOR Peruzzo et al. (2025) leveraging temporal redundancy to reduce the computational load in video pruning.

**Token Pruning in Large Vision-Language Models:** Token pruning in large vision-language models (LVLMs) is challenging due to the complex interplay between visual and textual inputs, with visual tokens often dominating input length and driving up inference costs. Early methods rely on attention scores for pruning Shang et al. (2024); Chen et al. (2024). PruMerge Shang et al. (2024) clusters and merges visual tokens based on attention sparsity in the vision encoder, while FastV Chen et al. (2024) prunes tokens in a specific LLM layer using attention magnitudes from earlier layers. However, attention-based pruning is suboptimal, particularly at high pruning ratios Guo et al. (2024). In other hand, Calibration-based methods offer another line of work, where pruning layers and/or ratios are determined by analyzing the LLM outputs for a calibration dataset Ye et al. (2025); Lin et al. (2025). For example, FitPrune Ye et al. (2025) calculates a pruning recipe based on the observed attention divergence before and after pruning. VTW Lin et al. (2025) argues that visual tokens can be entirely removed after a certain layer within LLM. To address the challenges of previous method, Diversity-aware methods such as DivPrune Alvar et al. (2025) ensure broad semantic coverage by maximizing the diversity of selected tokens. However, most existing approaches optimize either token importance or diversity in isolation, often relying on heuristic criteria. Our work addresses these limitations by formulating token pruning as a bi-criteria optimization problem that jointly balances token importance—measured via gradient-based saliency—and diversity using a Max-Min heuristic. This principled approach enables efficient and semantically rich token selection, substantially reducing inference costs in LVLMs without sacrificing accuracy.

## 3 METHOD

### 3.1 PRELIMINARY

**Large Vision-Language Models**: Let an input pair $(\mathbf{q}, \mathbf{I})$ denote the textual and visual inputs, respectively. The textual input $\mathbf{q}$ is tokenized into a sequence of $N$ tokens $\mathbf{T} = [\mathbf{t}_1, \ldots, \mathbf{t}_N]$, where each token $\mathbf{t}_i \in \mathbb{R}^d$ is a $d$-dimensional embedding. Similarly, the visual input $\mathbf{I}$ is processed by a vision encoder (e.g., ViT Dosovitskiy et al. (2021)) to extract features, which are then projected into the language embedding space. This yields a sequence of $M$ visual tokens $\mathbf{X} = [\mathbf{x}_1, \ldots, \mathbf{x}_M]$, where each $\mathbf{x}_j \in \mathbb{R}^d$.

The combined input to the large language model (LLM) is the concatenation of textual and visual tokens, $\mathbf{E} = [\mathbf{T}; \mathbf{X}] \in \mathbb{R}^{(N+M) \times d}$, where $[\cdot; \cdot]$ denotes concatenation along the sequence dimension. The LLM then autoregressively generates an output sequence $\mathcal{Y} = (\mathbf{y}_1, \ldots, \mathbf{y}_{\hat{N}})$, where the probability of generating the sequence is given by the chain rule:

$$P(\mathcal{Y} \mid \mathbf{E}) = \prod_{i=1}^{\hat{N}} P(\mathbf{y}_i \mid \mathbf{y}_{<i}, \mathbf{E}),\tag{1}$$

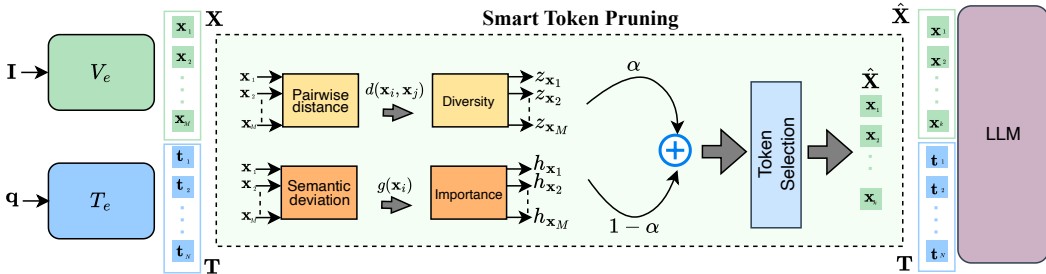

Figure 2: Given an image $\mathbf{I}$ and a query $\mathbf{q}$, we extract vision tokens $\mathbf{X}$ from a vision encoder $V_e$ and text tokens $\mathbf{T}$ from a text encoder $T_e$. STP prunes vision tokens based on diversity and importance scores. Specifically, each token is processed by diversity and importance modules, yielding scores $z_{\mathbf{x}i}$ and $h_{\mathbf{x}_i}$, which are then combined using a hyperparameter $\alpha$. The combined scores guide a token selection module that outputs a pruned set $\hat{\mathbf{X}}$. Finally, $\hat{\mathbf{X}}$ and $\mathbf{T}$ are concatenated and fed into an LLM to generate output tokens.

where $P(\cdot)$ is the model's output probability distribution, conditioned on the input $\mathbf{E}$ and the previously generated tokens $\mathbf{y}_{<i} = (\mathbf{y}_1, \ldots, \mathbf{y}_{i-1})$.

**Token Pruning**: In typical LVLMs Liu et al. (2023a), the number of visual tokens significantly exceeds the number of text tokens ($M \gg N$). This imbalance leads to significant memory and computational costs. To mitigate this, we propose pruning the visual token sequence by selecting a compact subset $\tilde{\mathbf{X}} \subset \mathbf{X}$ of size $k \ll M$, while preserving the model's output behavior. We define the token selection as a function $f : \mathbf{X} \to \tilde{\mathbf{X}}$. The objective is to find a function $f$ that minimizes the discrepancy between the output distributions of the original and pruned models:

$$\min_f \mathcal{L}\left(P(\mathcal{Y} \mid [\mathbf{T}; \mathbf{X}]), P(\mathcal{Y} \mid [\mathbf{T}; f(\mathbf{X})])\right) \quad \text{s.t.} \ |f(\mathbf{X})| = k, \tag{2}$$

where $\mathcal{L}(\cdot, \cdot)$ is a distance metric (e.g., KL divergence) and $|f(\mathbf{X})|$ denotes the number of tokens in the pruned sequence. The goal is to retain a minimal yet semantically rich set of visual tokens to enable efficient and accurate multimodal reasoning.

## 3.2 SMART TOKEN PRUNING

We propose STP, a novel visual token pruning framework that preserves tokens that are both semantically important and structurally diverse. Compared to previous methods that focus solely on either the importance of visual tokens or their structural diversity, STP formulates token selection as a dual-objective optimization problem, integrating *gradient-based saliency estimation* with a *diversity-aware dissimilarity criterion*. STP is fully plug-and-play, model-agnostic, and can be seamlessly applied to any pretrained LVLM model without requiring additional fine-tuning or calibration data. The pipeline of our proposed token pruning strategy, STP, is illustrated in Figure 2 and will be detailed in the following sections.

### 3.2.1 OPTIMIZATION OBJECTIVE:

To address the dual challenges of preserving semantic content and reducing redundancy, STP formulates visual token pruning as a joint optimization over two core principles: *diversity*, which ensures broad spatial and semantic coverage across the image, and *token-level importance*, which prioritizes the retention of tokens that are most critical to the model's final prediction. By jointly considering both factors, STP aims to maintain the expressiveness of the input representation while significantly reducing computational overhead.

We formalize the token selection process as a subset maximization problem over a combined diversity–importance objective:

$$S^* = \arg\max_{\substack{S \subset V \\ |S|=k}} \left[(1-\alpha) \cdot \mathcal{D}(S) + \alpha \cdot \mathcal{I}(S)\right], \tag{3}$$

where $\alpha \in [0, 1]$ is a balancing parameter that controls the trade-off between the diversity and importance terms. The diversity score $\mathcal{D}(S)$ encourages selection of tokens that are mutually dissimilar

in feature space, promoting coverage of distinct spatial and semantic regions. The importance score $\mathcal{I}(S)$ captures the aggregate influence of the selected tokens on the model's output, computed using task-aware saliency signals such as gradient-based relevance and activation strength.

### 3.2.2 DIVERSITY TERM:

To ensure semantic coverage, we define a diversity objective that selects tokens maximally dissimilar in feature space, quantified as:

$$\mathcal{D}(S) = \min_{\substack{\mathbf{x}_i, \mathbf{x}_j \in S \\ i \neq j}} d(\mathbf{x}_i, \mathbf{x}_j), \tag{4}$$

where $d(\mathbf{x}_i, \mathbf{x}_j) = 1 - \frac{\mathbf{x}_i^\top \mathbf{x}_j}{\|\mathbf{x}_i\| \|\mathbf{x}_j\|}$ represents the cosine distance between tokens $\mathbf{x}_i$ and $\mathbf{x}_j$. This *dissimilarity* criterion ensures that the selected tokens span a wide region of the feature space, discouraging redundancy and promoting semantic coverage.

*Diversity scoring:* At the conclusion of this stage, we compute a per-token diversity score for each vision token $z_{\mathbf{x}_i}$:

$$z_{\mathbf{x}_i} = \min_{\mathbf{x}_j \in S \setminus \{\mathbf{x}_i\}} d(\mathbf{x}_i, \mathbf{x}_j), \tag{5}$$

which measures how much unique semantic content token $i$ contributes relative to others. These diversity scores are later combined with importance scores in our joint token pruning framework, guiding the retention of tokens that are both semantically informative and non-redundant.

### 3.2.3 IMPORTANCE VIA GRADIENT-BASED SALIENCY:

To complement the diversity objective, we define an importance term $\mathcal{I}(S)$ that quantifies the collective saliency of a token subset $S$. This term is designed to measure the total contribution of the selected tokens to the model's representational capacity. We formulate the aggregate importance as the sum of the individual saliency scores of the tokens within the subset:

$$\mathcal{I}(S) = \sum_{\mathbf{x}_i \in S} h(\mathbf{x}_i), \tag{6}$$

where $h(\mathbf{x}_i)$ is a per-token saliency score reflecting the significance of token $\mathbf{x}_i$. A higher $\mathcal{I}(S)$ indicates that the subset $S$ contains tokens that are, in aggregate, more critical for the model's task. The following sections detail computing the individual saliency score $h(\mathbf{x}_i)$ for each token.

**Saliency Framework:** Identifying which tokens contribute most to an image's representation is essential for effective pruning and downstream reasoning. We propose a *gradient-based saliency framework* that scores tokens in two stages: (1) measuring each token's deviation from the global semantic distribution, and (2) weighting this deviation by its strongest activation response to prioritize semantically distinctive and highly informative tokens.

*Semantic Deviation:* To establish a semantic reference point, we compute the global mean and variance of the token representations:

$$\boldsymbol{\mu} = \frac{1}{M} \sum_{i=1}^{M} \mathbf{x}_i, \qquad \boldsymbol{\sigma} = \sqrt{\frac{1}{M} \sum_{i=1}^{M} (\mathbf{x}_i - \boldsymbol{\mu})^2}. \tag{7}$$

Here, $\boldsymbol{\mu}$ captures the semantic "center" of the representation, while $\boldsymbol{\sigma}$ characterizes the spread of tokens. Each token is then normalized with respect to these statistics to reveal its contextual deviation:

$$\tilde{\mathbf{x}}_i = \frac{\mathbf{x}_i - \boldsymbol{\mu}}{\boldsymbol{\sigma} + \epsilon}. \tag{8}$$

This normalization yields a gradient-like signal emphasizing tokens distinct from the overall feature distribution; we then use each token's L2 norm to quantify its deviation:

$$g(\mathbf{x}_i) = \|\tilde{\mathbf{x}}_i\|_2. \tag{9}$$

Tokens with higher $g(\mathbf{x}_i)$ values are interpreted as carrying more unique or discriminative information relative to the global feature space.

*Importance Score Calculation:* To further refine our understanding of token relevance, we incorporate activation strength into the saliency measure. Specifically, we compute the final per-token

importance score $h(\mathbf{x}_i)$ by weighting the gradient-based saliency $g(\mathbf{x}_i)$ by the token's most dominant feature activation:

$$h(\mathbf{x}_i) = g(\mathbf{x}_i) \cdot \max_j |\mathbf{x}_{i,j}|. \tag{10}$$

This weighting highlights tokens that both diverge from the global context and show strong activations, emphasizing those most informative for downstream tasks.

### 3.2.4 TOKEN SELECTION:

At this stage, for each vision token, we combine the diversity score $z_{\mathbf{x}_i}$ and the importance score $h_{\mathbf{x}_i}$ using a weighting coefficient $\alpha \in [0, 1]$:

$$u_{\mathbf{x}_i} = \alpha\, z_{\mathbf{x}_i} + (1 - \alpha)\, h_{\mathbf{x}_i}. \tag{11}$$

Finally, the unified scores $u_{\mathbf{x}_i}$ are passed to the token selection module, which ranks vision tokens and retains the top-$k$ based on a predefined pruning ratio. This lightweight ranking requires only one pass after score computation, yielding a condensed yet representative token subset for subsequent model layers. In the next section, we provide experimental results highlighting the benefits of STP.

## 4 EXPERIMENTS

**Baselines and Models:** We compare STP against three baselines: FastV Chen et al. (2024), VTW Lin et al. (2025), and DivPrune Alvar et al. (2025), which represent recent state-of-the-art approaches for visual token pruning or selection. To evaluate the robustness and generalizability of our approach, we benchmark STP and all baselines across multiple popular Large Multimodal Models (LMMs), including LLaVA 1.5-7B Liu et al. (2023a), LLaVA 1.5-13B Liu et al. (2023a), and LLaVA 1.6-7B (also referred to as LLaVA-NeXT Liu et al. (2024a)). These models vary in size, architecture, and visual token processing strategies, making them suitable testbeds for assessing performance across diverse configurations. Specifically, all LMMs use a CLIP-based vision encoder Radford et al. (2021) to extract visual features. LLaVA 1.5 encodes each image into a fixed-length sequence of 576 visual tokens, while LLaVA 1.6 adopts a more flexible strategy, converting images into variable-length token sequences that are 3–5× longer, by using adaptive patching. For each model-task pair, we include only the relevant baselines that are compatible or directly applicable. This ensures fair comparisons while highlighting the adaptability and performance gains of our method across varying model sizes and token granularities.

**Datasets, Tasks, and Metrics:** We evaluate our method on a diverse suite of 11 image-language datasets designed to assess multimodal reasoning and understanding. These datasets cover a broad spectrum of tasks, including image captioning, multiple-choice question answering (QA), and open-ended QA grounded in both text and image/video inputs. Following established benchmarks, we adopt CIDEr Vedantam et al. (2015) for captioning evaluation, and use Exact Match (EM), Accuracy (Acc), F1, Perception Score (P-score) Yin et al. (2024), and GPTScore Fu et al. (2023) for QA tasks. For open-ended QA, we further incorporate Wu-Palmer Similarity (WUPS) Wu & Palmer (1994) to capture semantic alignment. Higher values in all metrics indicate better performance. All experiments were conducted using 8 NVIDIA A100 GPUs (80GB VRAM). We utilize the `lmmsevals` package Zhang et al. (2024) to ensure standardised evaluation across all models and baselines. Unless otherwise specified, a batch size of 1 is used for all reported results.

### 4.1 VISION-LANGUAGE UNDERSTANDING

In this section, we evaluate our proposed method, STP, against baseline methods across a range of image-language understanding tasks, including open-ended and closed-ended question answering, visual reasoning, and image captioning. Specifically, we use the following datasets: ScienceQA-IMG (SQA) Lu et al. (2022), POPE Li et al. (2023c), MME Yin et al. (2024), MMB Liu et al. (2024b), GQA Hudson & Manning (2019), MMMU Yue et al. (2024), Flickr30k Plummer et al. (2015), SeedBench (SEEDB) Li et al. (2023a), Nocaps Agrawal et al. (2019), OKVQA Marino et al. (2019), and COCO-2017 Lin et al. (2014). All the experiments are shown in Table 1.

**LLaVA 1.5-7B.** For the 7B variant of LLaVA 1.5, our method achieves a strong efficiency–accuracy trade-off, operating at only 15.63% of the original computational cost while maintaining consistent

| Method | TFLOP (ratio %) | COCO CIDEr | Flickr CIDEr | GQA EM | MMB Acc | MME P-score | MMMU Acc | Nocaps CIDEr | OKVQA EM | POPE F1 | SQA EM | SEEDB Acc |
|---|---|---|---|---|---|---|---|---|---|---|---|---|
| **LLaVA 1.5-7B** | | | | | | | | | | | | |
| Original | 3.228 (100.00) | 1.10 | 0.75 | 61.96 | 64.09 | 1506 | 36.44 | 1.06 | 53.39 | 85.84 | 69.41 | 66.17 |
| VTW | 0.603 (18.46) | 0.05 | 0.03 | 38.94 | 21.31 | 681 | 32.60 | 0.03 | 18.64 | 25.35 | 65.29 | 36.13 |
| FastV | 0.514 (15.69) | 0.06 | 0.03 | 38.73 | 20.62 | 696 | 32.00 | 0.04 | 18.32 | 32.84 | 65.15 | 35.69 |
| Divprune | 0.512 (15.63) | 0.96 | 0.62 | 56.85 | 59.19 | **1328** | **35.89** | 0.92 | 46.98 | **86.02** | 68.27 | 59.47 |
| **Ours** | 0.512 (15.63) | **0.97** | **0.64** | **57.64** | **59.45** | 1306 | 34.33 | **0.93** | **47.84** | 85.25 | **68.37** | **59.64** |
| **LLaVA 1.5-13B** | | | | | | | | | | | | |
| Original | 6.281 (100.00) | 1.16 | 0.80 | 63.33 | 68.64 | 1522 | 35.67 | 1.09 | 58.28 | 85.99 | 72.88 | 66.82 |
| VTW | 1.030 (16.16) | 0.08 | 0.05 | 39.71 | 21.91 | 622 | 32.10 | 0.05 | 22.49 | 0.40 | 66.24 | 38.59 |
| FastV | 1.003 (15.73) | 0.38 | 0.18 | 44.98 | 37.80 | 942 | 35.11 | 0.33 | 32.14 | 30.02 | 69.96 | 44.95 |
| Divprune | 1.002 (15.71) | 1.00 | 0.66 | 57.29 | 63.40 | 1407 | 34.89 | 0.95 | **53.29** | 83.43 | **72.34** | **62.04** |
| **Ours** | 1.002 (15.71) | **1.02** | **0.67** | **57.64** | **64.17** | **1438** | **36.22** | **0.97** | 53.10 | **84.68** | 71.44 | 61.78 |
| **LLaVA 1.6-7B** | | | | | | | | | | | | |
| Original | 11.849 (100.00) | 1.00 | 0.68 | 64.28 | 67.01 | 1520 | 36.44 | 0.88 | 44.20 | 86.38 | 70.15 | 70.16 |
| VTW | 1.318 (11.23) | 0.06 | 0.03 | 38.62 | 19.76 | 606 | 31.30 | 0.03 | 8.66 | 7.13 | 65.74 | 37.48 |
| FastV | 1.327 (11.30) | 0.06 | 0.03 | 38.79 | 20.36 | 619 | 32.56 | 0.04 | 8.80 | 7.78 | 65.49 | 37.62 |
| Divprune | 1.266 (10.79) | 0.89 | 0.61 | 58.69 | 63.49 | 1362 | **37.11** | 0.76 | 41.92 | 82.97 | 68.57 | 64.11 |
| **Ours** | 1.266 (10.79) | **0.9** | **0.62** | **59.77** | 63.83 | **1364** | 37.00 | **0.78** | **42.17** | **84.37** | 67.48 | **64.31** |

Table 1: Comparison results of our method with multiple baselines on image understanding tasks.

performance across all benchmarks. Unlike VTW and FastV, which show large drops on reasoning-heavy tasks (e.g., GQA: 38.94/38.73 vs. 57.64; MMB: 21.31/20.62 vs. 59.45), our approach preserves visual–linguistic signals more effectively. It also surpasses DivPrune under similar FLOP constraints with gains on COCO (0.97 vs. 0.96), Flickr (0.64 vs. 0.62), OKVQA (47.84 vs. 46.98), and SEED-Bench (59.64 vs. 59.47). These results demonstrate our ability to retain semantically informative visual tokens under extreme computational efficiency *without additional fine-tuning*, highlighting robustness across tasks and scalability to larger LVLMs and real-world scenarios.

**LLaVA 1.5-13B.** On the 13B variant of LLaVA 1.5, our method retains its efficiency advantage (15.71% of original FLOPs) while outperforming all baselines on most tasks. VTW and FastV again show sharp drops on key benchmarks (e.g., COCO: 0.08/0.38 vs. 1.02; GQA: 39.71/44.98 vs. 57.64), reflecting limited semantic retention under pruning. While DivPrune performs competitively, our method yields further gains on reasoning-intensive tasks such as MMB (64.17 vs. 63.40), MMMU (36.22 vs. 34.89), and POPE (84.68 vs. 83.43). These consistent improvements indicate that our pruning strategy scales effectively with larger model capacities, leveraging richer feature hierarchies *without retraining or calibration*. Note that, our approach maintains performance close to the original model while reducing compute by over 84%, showing it not only generalizes across model sizes but also benefits from the expanded representational space of larger LVLMs.

**LLaVA 1.6-7B.** Applied to the latest LLaVA 1.6-7B, our method sustains superior performance under stricter compute limits, operating at only 10.79% of original FLOPs. VTW and FastV degrade sharply (e.g., OKVQA: 8.66/8.80 vs. 42.17; POPE: 7.13/7.78 vs. 84.37), showing poor generalization in highly compressed regimes. DivPrune remains competitive, yet our approach consistently surpasses it with gains on GQA (59.77 vs. 58.69), OKVQA (42.17 vs. 41.92), and POPE (84.37 vs. 82.97), while matching or slightly exceeding performance elsewhere. These results show our method generalizes to newer architectures, preserving low-level visual grounding and high-level multimodal reasoning even under aggressive pruning. Crucially, its robustness under extreme FLOP reductions highlights seamless adaptation to evolving LVLMs *without architectural changes*, positioning it as a future-proof solution for next-generation multimodal systems with rising efficiency demands.

## 4.2 TIME AND EFFICIENCY ANALYSIS

We evaluate STP across three LLaVA variants (LLaVA-1.6-7B, LLaVA-1.5-7B, and LLaVA-1.5-13B) to demonstrate its effectiveness and generalizability. As shown in Figure 3, STP consistently improves memory efficiency and inference speed. The largest gains occur on LLaVA-1.6-7B, with 13.6% memory reduction (15.75 GB to 13.61 GB) and 34.1% faster inference (483.28 ms to 318.48 ms). LLaVA-1.5-13B achieves 3.3% memory and 20.4% latency improvements, while

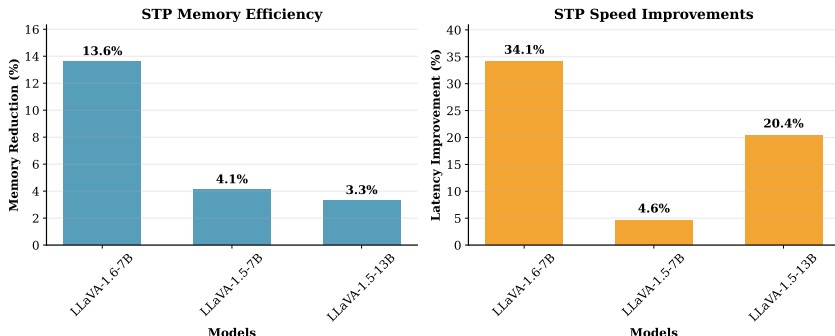

Figure 3: STP performance across LLaVA variants. Left: memory reduction; Right: latency improvement. Consistent gains are observed for all models, with LLaVA-1.6-7B achieving the largest improvements (13.6% memory reduction, 34.1% faster inference).

LLaVA-1.5-7B shows 4.1% memory and 4.6% speed gains. On average, STP reduces memory by 7.0% and latency by 19.7% across all models, demonstrating robust generalization without performance loss. These results highlight STP's value for edge deployment, enabling faster, more efficient large-vision-language models in resource-constrained settings.

## 4.3 ABLATION STUDY

**The impact of $\alpha$ in terms of diversity and importance:** To assess the generalization capability of STP across different task categories (see Figure 4), we analyze performance patterns grouped by task type: Visual Question Answering (VQA: GQA, OK-VQA, ScienceQA), Multimodal Reasoning (MMBench, MMMU, SeedBench), and Specialized Evaluation (POPE). Our analysis reveals that all three categories maintain consistently high performance retention (94–100%) across the entire range of $\alpha$ values (0.20–0.37), demonstrating the method's robustness across diverse task domains. The VQA category shows the most stable performance with minimal variance, while the Reasoning category exhibits slightly higher variability, potentially due to the more complex cognitive demands of these tasks. The Specialized category, represented by POPE (hallucination detection), maintains

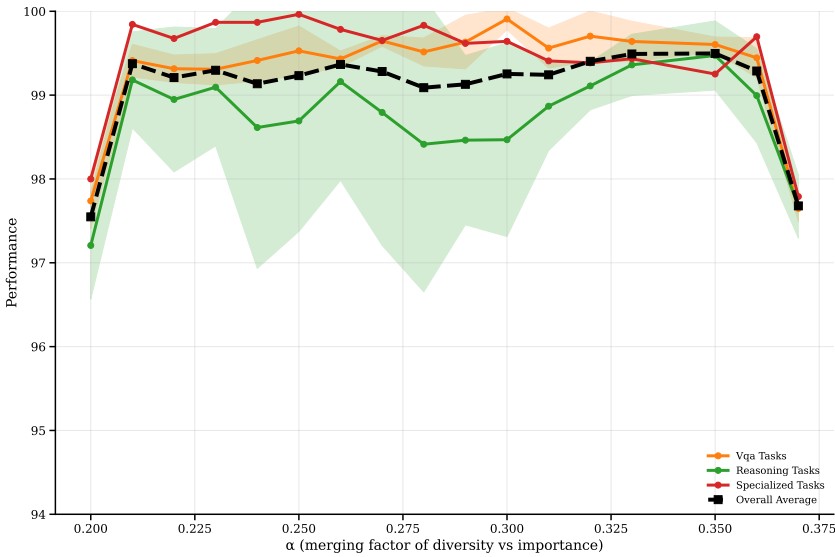

Figure 4: Task category robustness showing performance by type (VQA, Reasoning, Specialized) as $\alpha$ varies. Category averages (colored lines with shaded regions) and overall average (black dashed) indicate consistent performance, demonstrating robust pruning.

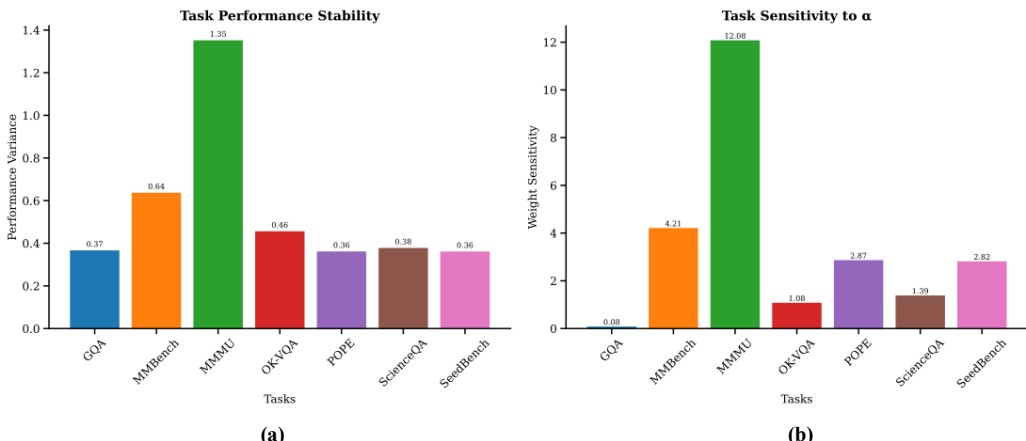

(a)                          (b)

Figure 5: STP robustness across varying $\alpha$ on seven benchmarks. Performance stays above 94% even under extreme pruning, demonstrating stable diversity–importance trade-offs. Reasoning-heavy tasks show slightly more variance, while VQA and specialized tasks maintain near-flat trends, underscoring STP's consistency.

stable performance despite its unique evaluation criteria. The overall average performance across all categories follows a similar trend to individual tasks, confirming that DivPrune preserves model capabilities uniformly across different types of vision-language understanding tasks, regardless of their specific evaluation metrics or cognitive requirements. Interestingly, this consistency suggests that $\alpha$ serves as a largely architecture-agnostic balancing knob, enabling a smooth trade-off between diversity and importance without destabilizing performance. Moreover, the narrow optimal range of $\alpha$ values implies that STP requires minimal hyperparameter tuning, making it practical for deployment across heterogeneous multimodal pipelines.

**Robustness analysis:** We conduct a comprehensive sensitivity analysis to evaluate how the merging factor $\alpha$ (balancing diversity versus importance in STP) affects model performance across diverse vision-language tasks. As shown in Figure 5, our analysis focuses on seven accuracy-based benchmarks including GQA, MMBench, MMMU, OK-VQA, POPE, ScienceQA, and SeedBench, representing visual question answering, multimodal reasoning, and specialized evaluation tasks. The results demonstrate that performance retention remains consistently high (94–100%) across all tasks as $\alpha$ varies from 0.20 to 0.37, indicating the robustness of our pruning approach. Notably, individual tasks exhibit distinct sensitivity patterns: reasoning tasks (MMBench, MMMU, SeedBench) show slightly higher variance compared to VQA tasks (GQA, OK-VQA, ScienceQA), while specialized tasks (POPE) maintain stable performance. The average performance across all tasks reveals a subtle but consistent trend, suggesting that $\alpha$ values between 0.25–0.31 provide an optimal balance between computational efficiency and performance retention, with minimal degradation even at the most aggressive pruning levels ($\alpha = 0.20$). This resilience shows that STP's weighting mechanism is inherently stable, ensuring predictable behavior even when $\alpha$ deviates from its optimal range.

## 5 CONCLUSION

In this work, we proposed a principled and efficient token pruning framework for Large Vision-Language Models (LVLMs) that balances the dual objectives of token importance and semantic diversity. By formulating the pruning process as a bi-criteria optimization problem and introducing a novel gradient-based saliency score, our method effectively identifies a compact subset of visual tokens that preserves essential multimodal information. Extensive experiments across 11 large-scale benchmarks demonstrate that our approach achieves significant reductions in inference time and memory usage, with minimal performance degradation. Our method not only advances the efficiency of LVLMs but also provides a generalizable strategy applicable to a wide range of multimodal architectures. Future work includes extending the framework to dynamic token selection during training, incorporating task-specific adaptive pruning, and exploring cross-modal token interactions for deeper reasoning.

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
