# OpenReview forum: "STP: Smart Token Pruning for Vision-Language Models: Balancing Importance and Diversity"
_ICLR.cc/2026/Conference — ICLR 2026 Conference Withdrawn Submission_

### Official Review · Reviewer_gxZk · 2025-10-27

**Soundness:** 2
**Presentation:** 2
**Contribution:** 1
**Rating:** 2
**Confidence:** 4

**Summary:**

This paper proposes **STP (Smart Token Pruning)**, a framework for efficient inference in large vision–language models (LVLMs). STP formulates token pruning as a **bi-criteria optimization** problem that jointly maximizes **semantic diversity** and **token importance** using a gradient-based saliency score. By adaptively selecting tokens based on this joint score, STP reduces computational and memory overhead while preserving visual-textual reasoning performance. Experiments on 11 benchmarks demonstrate that STP significantly improves efficiency without sacrificing accuracy, enabling scalable deployment of LVLMs.

**Strengths:**

1. The paper is clear and practical, addressing the trade-off between token importance and semantic diversity to improve efficiency in LVLMs. And the manuscript is explicit and well-organized.

2. Experimental validation is sufficient. The authors conduct comprehensive experiments on various tasks and show **improvements**, to validate the effectiveness of the method. Moreover, the ablation study is detailed, **particularly in the efficiency analysis section**.

**Weaknesses:**

1. This paper feels like a combination of three existing ideas, and I find it lacks a clear motivation.

2. It would be helpful to include comparisons with methods such as VisionZip [1] and SparseVLM [2] in the table 1.

3. Table 1 lists the performance on three models, but it does not include results for additional token settings.

[1] VisionZip, CVPR 2025.

[2] SparseVLM, ICML 2025.

**Questions:**

1. I would like to see a visualization of actual token pruning.

---

### Official Review · Reviewer_uFJD · 2025-10-27

**Soundness:** 3
**Presentation:** 3
**Contribution:** 2
**Rating:** 2
**Confidence:** 4

**Summary:**

Prior token-pruning methods typically trade off two needs: keeping the most informative tokens or maintaining semantic coverage. This paper presents STP, treats pruning as a bi-criteria problem, jointly promoting semantic diversity and token importance via a gradient-based saliency measure and an adaptive selection rule. On 11 benchmarks, it notably reduces computation and memory while keeping accuracy competitive, making LVLM inference more practical.

**Strengths:**

1. This paper proposed a method that balances semantic diversity and token importance, which does not require training or rebuilding the model architecture.

2. This paper not only reports TFLOPs, but also memory reduction and latency improvement in real-time running, which makes it easy to deploy.

3. The paper is well written and easy to read.

**Weaknesses:**

1. Generalization and robustness on other model architectures have not yet been verified: The paper lacks experiments on different models; they only conduct experiments primarily on the LLaVA family; effectiveness on other architectures (e.g., Qwen-VL[1], InstructBLIP[2]) remains to be validated.

2. Baseline selection is not enough: The comparison with existing methods is not entirely up-to-date. Therefore, more methods should be compared, for example, the VisPruner[3], DART[4], CDPruner[5], and so on.

3. The comparison with DivPrune is relatively small: The gap between the proposed method and DivPrune appears relatively small when FLOPs are similar, and in some datasets, their performance is comparable. It might strengthen the work to include a more systematic evaluation under higher pruning ratios or more demanding scenarios, such as long videos or high-resolution inputs.

[1] Jinze Bai, Shuai Bai, Shusheng Yang, Shijie Wang, Sinan Tan, Peng Wang, Junyang Lin, Chang Zhou, and Jingren Zhou. Qwen-vl: A versatile vision-language model for understanding, localization, text reading, and beyond. arXiv preprint arXiv:2308.12966, 2023.

[2] Wenliang Dai and Junnan Li and Dongxu Li and Anthony Meng Huat Tiong and Junqi Zhao and Weisheng Wang and Boyang Li and Pascale Fung and Steven Hoi. InstructBLIP: Towards General-purpose Vision-Language Models with Instruction Tuning. ArXiv, 2023a.

[3] Zhang, Q., Cheng, A., Lu, M., Zhuo, Z., Wang, M., Cao, J., ... & Zhang, S. (2024). [CLS] Attention is All You Need for Training-Free Visual Token Pruning: Make VLM Inference Faster. ICCV, 2025.

[4] Wen, Z., Gao, Y., Wang, S., Zhang, J., Zhang, Q., Li, W., ... & Zhang, L. (2025). Stop looking for important tokens in multimodal language models: Duplication matters more. EMNLP, 2025.

[5] Zhang, Q., Liu, M., Li, L., Lu, M., Zhang, Y., Pan, J., ... & Zhang, S. (2025). Beyond Attention or Similarity: Maximizing Conditional Diversity for Token Pruning in MLLMs. NeurIPS, 2025.

**Questions:**

1. Could you report memory reduction and latency improvement compared with other baselines? A side-by-side summary under matched settings would help compare the method's improvement.

---

### Official Review · Reviewer_g1Hy · 2025-10-30

**Soundness:** 3
**Presentation:** 3
**Contribution:** 2
**Rating:** 4
**Confidence:** 5

**Summary:**

This paper proposes STP (Smart Token Pruning), a training-free, bi-criteria token-selection framework for efficient inference in large vision-language models (LVLMs). STP formulates pruning as a joint optimization over token importance and semantic diversity, introducing: a gradient-based saliency score combining feature sensitivity and activation strength, and a Max–Min diversity objective to preserve broad visual coverage. The method is model-agnostic and plug-and-play, requiring no retraining. Experiments on 11 benchmarks with LLaVA-1.5 (7B/13B) and LLaVA-1.6 (7B) show that STP matches or slightly outperforms DivPrune, FastV, and VTW, retaining 96–100 % accuracy while reducing computation to ~10–16 % of baseline FLOPs. It further improves latency (≈ 20–34 %) and memory usage (≈ 7 %).

**Strengths:**

1. Joint diversity–importance formulation mitigates redundancy while retaining key evidence.
2. Model-agnostic and training-free: Works across LVLM architectures without retraining or calibration data.
3. Gradient-based saliency: Captures task-relevant sensitivity better than attention-only metrics.

**Weaknesses:**

1. Conceptually close to DivPrune; adds a simple gradient-importance term rather than a new pruning paradigm. In addition, it only marginally outperforms DivPrune in most experiments with a single run experiment (lack of multiple run info).
2. Approximate saliency: Uses activation deviation instead of true gradients, so importance is heuristic.
3. Manual \alpha tuning: Trade-off coefficient is fixed (≈ 0.28) and not data-adaptive.
4. Single-stage pruning: Lacks layer-wise or task-aware adaptation present in FitPrune/VTW.

**Questions:**

1. Can \alpha be made adaptive to input complexity or gradient variance?
2. How well does STP generalize to multi-image or video inputs?
3. Could token-importance visualization shed light on interpretability?
4. What is the quantitative overhead of computing saliency versus attention-based scores?

---

### Official Review · Reviewer_DfY9 · 2025-10-31

**Soundness:** 3
**Presentation:** 3
**Contribution:** 3
**Rating:** 4
**Confidence:** 3

**Summary:**

This paper presents STP (Smart Token Pruning), a framework for improving the efficiency of Large Vision-Language Models (LVLMs). The authors formulate token pruning as a bi-criteria optimization problem that balances semantic diversity and token importance. A new gradient-based saliency score integrates feature sensitivity and activation strength to estimate importance, while a Max–Min diversity strategy encourages semantic coverage. Experiments on 11 multimodal benchmarks (e.g., POPE, MME, GQA, MMMU, COCO, etc.) show that STP achieves over 80–90% FLOP reduction while maintaining comparable or even slightly better performance than baselines such as FastV, VTW, and DivPrune.

**Strengths:**

1. Comprehensive and systematic evaluation across multiple LVLMs and datasets.
2. Strong efficiency gains (10–15% of original FLOPs) with minimal performance degradation.
3. Ablation and sensitivity analyses provide evidence of robustness and hyperparameter stability.
4. The method is training-free and model-agnostic, improving its applicability.

**Weaknesses:**

1. The method mainly combines two existing heuristics (diversity + importance) without deep theoretical grounding.
2. Missing discussion on why the gradient-based saliency is superior to existing attention-based or similarity-based metrics.
3. No analysis on token distribution or interpretability of pruned tokens.
4. The comparisons focus primarily on a limited set of baselines (FastV, DivPrune, VTW); missing representative token compression works (e.g., ToMe[1], FitPrune[2], DART[3], AiM[4]).
5. Minor clarity issues in formulas and parameter explanations.

[1] Bolya, Daniel, et al. "Token merging: Your vit but faster." arXiv preprint arXiv:2210.09461 (2022).
[2] Ye, Weihao, et al. "Fit and prune: Fast and training-free visual token pruning for multi-modal large language models." Proceedings of the AAAI Conference on Artificial Intelligence. Vol. 39. No. 21. 2025.
[3] Wen, Zichen, et al. "Stop looking for important tokens in multimodal language models: Duplication matters more." arXiv preprint arXiv:2502.11494 (2025).
[4] Zhong, Yiwu, et al. "Aim: Adaptive inference of multi-modal llms via token merging and pruning." Proceedings of the IEEE/CVF International Conference on Computer Vision. 2025.

**Questions:**

1. How sensitive is STP to the choice of the balancing parameter α across unseen models or datasets?
2. Have you considered pruning textual tokens or performing joint multimodal token selection?
3. Could gradient-based saliency be unstable under low batch sizes or noisy gradients?
4. What is the real-time latency improvement on edge devices beyond FLOP reduction?
5. Is it possible to integrate STP into training-time pruning for further acceleration?
6. To verify the true effectiveness of the method, it is recommended to conduct an evaluation within this framework [1]

[1] Liao, Chenfei, et al. "Are We Using the Right Benchmark: An Evaluation Framework for Visual Token Compression Methods." arXiv preprint arXiv:2510.07143 (2025).

---

### Note · Authors · 2025-11-17

I have read and agree with the venue's withdrawal policy on behalf of myself and my co-authors.